# The Impact of MRSA Colonization on Healthcare-Associated Infections in Long-Term Care Facility Residents: A Whole-Genome Sequencing-Based Study

**DOI:** 10.3390/microorganisms11122842

**Published:** 2023-11-23

**Authors:** Manuel Callejón Fernández, Rossana Abreu Rodríguez, Ángeles Arias, Armando Aguirre-Jaime, María Beatriz Castro Hernández, María José Ramos Real, Yanet Pedroso Fernández, María Lecuona

**Affiliations:** 1Microbiology and Infection Control Service, University Hospital of the Canary Island (HUC), 38320 La Laguna, Spain; mbcasher@gobiernodecanarias.org (M.B.C.H.); mjoseramosreal@hotmail.com (M.J.R.R.); yanetcub@yahoo.es (Y.P.F.); mlecuona2005@yahoo.es (M.L.); 2Doctoral Program in Medical and Pharmaceutical Sciences, Development and Quality of Life, University of La Laguna (ULL), Campus de Ofra s/n, 38071 Santa Cruz de Tenerife, Spain; 3Department of Preventive Medicine and Public Health, University of La Laguna (ULL), Campus de Ofra s/n, 38200 Santa Cruz de Tenerife, Spain; rossana_abreu@hotmail.com (R.A.R.); angarias@ull.edu.es (Á.A.); 4Institute of Care Research, Nurses Association of Santa Cruz de Tenerife, C. San Martin, 63, 38001 Santa Cruz de Tenerife, Spain; armagujai@gmail.com

**Keywords:** MRSA, long-term care facilities, colonization, infection, prevalence, whole-genome sequencing

## Abstract

Methicillin-resistant *Staphylococcus aureus* (MRSA) colonization has been considered a risk factor for the development of infection, however, there are no studies that have compared the colonizing and infecting strains using whole-genome sequencing (WGS). The aim of this study is to determine the prevalence of and risk factors for MRSA colonization among long-term care facilities (LTCF) residents of Tenerife (Spain), and to analyze the epidemiological relationship between the colonizing and infecting strains using WGS. A point-prevalence study was carried out at 14 LTCFs in Tenerife from October 2020 to May 2021. Nasal swabs were cultured for MRSA. Colonized residents were followed up for two years. A phylogenetic comparison between colonization and infection strains was performed using WGS. A total of 764 residents were included. The prevalence of colonization by MRSA was 28.1% (*n* = 215), of which 12 (5.6%) subsequently developed infection. A close genetic relationship between colonization and infection isolates was found in three of the four (75%) residents studied. Our study confirms that colonized residents can develop serious MRSA infections from the same nasal colonization strain. Given the high prevalence of MRSA colonization in these centers, it is necessary to implement strategies with preventive measures to avoid the development of infection and the transmission of MRSA.

## 1. Introduction

Methicillin-resistant *Staphylococcus aureus* (MRSA) represents a significant public health concern, and is responsible for large number of infections [1]. Its capacity to adapt and develop resistance to various antimicrobials may have facilitated its extensive dissemination worldwide. The aging of the population and the chronicity of certain diseases have resulted in an increased number of patients being placed in long-term care facilities (LTCF). It is known that these centers are significant reservoirs of this microorganism, serving as a potential conduit for the dissemination of MRSA in both hospitals and community settings [2,3]. Several studies have shown that MRSA colonization increases the risk of subsequent infection in hospital patients and LTCF residents [4,5,6,7], increasing morbimortality and healthcare costs. However, none of these studies have investigated whether the MRSA infection was caused by the same colonizing strain. The demonstration of the genotypic relationship between the strains would reinforce the importance of preventive strategies and propose early action in this group of patients, potentially reducing the number of MRSA infections.

On the other hand, hospitals and LTCFs are connected to residential settings through the transfer of residents or patients. The transmission of MRSA between these centers is not yet understood; some studies suggested that MRSA is predominantly imported from hospitals into LTCFs through patient transfers [8,9], however, other studies indicate that the high prevalence of MRSA colonization in LTCFs may be attributed to the cross-transmission of MRSA inside LTCFs [10,11]. Knowledge about the epidemiology of MRSA and the relationship between hospitals and LTCFs is an important public health issue, so more studies are necessary to help clarify the origin and transmission mechanism of MRSA.

Whole-genome sequencing (WGS) could hold the key to solving these issues. This technique allows us to determine the epidemiological relationship between the LTCFs and hospital strains, as well as between the nasal colonizing and infecting strains.

The aim of this study is to investigate the prevalence and risk factors associated with MRSA colonization among LTCF residents in the north of Tenerife (Spain) and to analyze the epidemiological relationship between colonizing and infecting strains using WGS.

## 2. Materials and Methods

### 2.1. Study Design

A multicenter point-prevalence study was conducted between October 2020 and May 2021, carried out in 14 LTFCs located in the north of Tenerife (Spain), which belonged to the reference area of our hospital. In total, 26 centers belonging to our reference area were contacted through the IISS (insular institute of social and socio-health care), with a total of 14 centers agreeing to participate (11 public centers and 3 private centers). The residents of these centers are people, mostly elderly, who require health care due to chronic illnesses or because they have a high degree of dependency. Informed consent was obtained from the patients or, in the case of an impaired mental state, from their designated family representative. Following the acquisition of informed consent, nasal samples were collected from all residents. The sociodemographic and clinical data of the residents were assessed using a questionnaire. None of the participating LTFCs had a specific protocol for monitoring and preventing the transmission of multi-drug resistant organisms (MDRO). A resident colonized by MRSA was categorized as a case, while a control was defined as an individual not colonized by MRSA. MRSA-colonized residents were followed over a two-year period to determine their hospital admissions, cultures and subsequent infections by MRSA. Infections and colonizations were classified according to Centers for Disease Control and Prevention’s criteria [12]. All data, both from medical history and microbiological cultures, were collected in a joint database to allow for statistical analysis. For residents who were later admitted to the hospital and had an MRSA infection, a phylogenetic analysis of both previous nasal colonization and infection MRSA strains was performed.

### 2.2. Microbiological Methods

All samples were analyzed in the Microbiology Service at the Hospital Universitario de Canarias, a reference hospital designated for the northern region of Tenerife. Nasal swabs were cultured directly on ChromID^®^ MRSA SMART (bioMérieux, Marcy l´Etoile, France) and in Brain-Heart Infusion Broth (bioMérieux). Following a 24 h incubation period, they were then reseeded in MRSA chromogenic medium. Colonies exhibiting characteristics suggestive of MRSA were confirmed using a PBP2A SA colony culture test (AlereTM Scarborough, ME, USA). Mupirocin and fusidic acid susceptibility were studied via disc diffusion on Mueller Hinton E agar (MHE, bioMérieux). Methicillin resistance genes (mecA, mecC) and the Panton-Valentine leucocidin (PVL) gene in *S. aureus* were identified using the eazyplex^®^ MRSAplus isothermal amplification system (Amplex Diagnostics, Giessen, Germany). All strains, both those from the study and from hospital clinical samples, were stored in a freezer at −80 °C, in order to be able to use them later for genomic analysis. Whole-genome sequencing (WGS) was performed using the MiSeqTM platform (Illumina Inc., San Diego, CA, USA). The methodology used for WGS was with Illumina reagents from their WGS protocol: “Whole genome sequencing with Illumina DNA prep technology”. The analysis of resistance phenotypes: Software 4.4.1 used was the Center for Genomic Epidemiology’s (CDC) (https://www.genomicepidemiology.org/, accessed on 15 March 2023). Phylogenetic analysis was performed using NDtree2.1 (https://cge.food.dtu.dk/services/NDtree, accessed on 15 March 2023) and sequence types (ST) were assigned using the multilocus sequence typing (MLST) of the total-genome-sequenced bacteria database (https://pubmlst.org, accessed on 15 March 2023). Finally, the analysis of the antibiotic resistance phenotype was carried out using Resfinder-4.4.1 (http://genepi.food.dtu.dk/resfinder, accessed on 4 April 2023).

### 2.3. Statistical Analysis

The samples collected, comprising 215 cases and 549 controls, conferred a study power of 90%, which made it possible to detect a significant difference between cases and controls, with relative frequency disparities for the nominal variables of at least 20%, or three-year distinctions in age and a range of days from 0 to 5 for the length of hospital stay. These evaluations were performed using two-sided hypothesis tests with a level of statistical significance set at *p* ≤ 0.05. The characteristics of the sample as a whole are reported by summarizing its nominal variables with the frequency (relative frequency) of its component categories, and those of the numerical scale with the mean (P5–P95), given its distance from a normal probability distribution, verified with the Kolmogorov–Smirnov test. Nominal variables for cases and controls are compared with Pearson’s chi2 test or Fisher’s exact test. Numerical scale comparisons were made using the Mann–Whitney U test. All hypothesis contrast tests were conducted as two-sided tests with a statistical significance level set at *p* ≤ 0.05. The calculations for these analyses were performed using the statistical package for data processing, SPSS 25.0™, developed by IBM Co.^®^ (IBM –SPSS Inc., Armonk, NY, USA).

### 2.4. Ethics Statement

This study was conducted with the authorization of the Institutional Review Board at the Hospital Universitario de Canarias (Tenerife, Spain), under the reference code CHUC_2019_91.

## 3. Results

A total of 764 residents from 14 LTFCs (10 public and 4 private) were included in the study (Table 1). A total of 215 (28.1%) were colonized by MRSA, of which 17 strains of MRSA (7.9%) were resistant to mupirocin, 9 (4.2%) were resistant to fusidic acid and 4 (1.9%) were resistant to both. The resistance mechanism for all strains of MRSA was mecA. Additionally, one of the isolated strains was positive for Panton-Valentine leucocidin (PVL) gen. In Table 1, we can observe the degree of participation and the prevalence of MRSA for each LTCF. In the univariate analysis, there were no factors significantly associated with MRSA colonization, except the factor of having had a previous MRSA colonization/infection (Table 2).

Among the 215 MRSA carriers, 90 (41.9%) had a subsequent admission to the hospital for medical care, of which 23 (25.5%) residents remained carriers of MRSA and 11 (12.2%) were negative for nasal MRSA screening. The remaining 56 (62.2%) residents were not screened for MRSA by the active surveillance of our hospital because they did not meet the criteria. Most of these residents had a brief admission (1 or 2 days) to the emergency department, while others were admitted to services with a notably low prevalence of MRSA. In both scenarios, our hospital does not conduct active surveillance due to cost-effectiveness considerations.

During the two years of follow-up for the MRSA carriers, 12 (5.6%) were diagnosed with an MRSA infection via culture, of which 7 were diagnosed on an outpatient basis and 5 during hospital admission. The time elapsed between the nasal MRSA collection in our study and the development of the infection was 2 to 16 months. All infections diagnosed on an outpatient basis were skin and soft tissue infections and in these residents it was not possible to compare colonization/infection isolates because, in our hospital routine, the strains of patients who are not hospitalized are not kept.

On the other hand, in the hospital, two bacteremia and three skin and soft tissue infections were diagnosed. One of the bacteremia infections was classified as a central venous catheter-associated hospital infection and the patient died a few days later. MRSA was first isolated from a skin and soft tissue ulcer exudate sample, and then from the central venous catheter and blood cultures. This patient was positive for nasal MRSA at hospital admission screening and was treated with 2% mupirocin for 5 days, and was negative in a control test at day 8. However, in this case, decolonization did not prevent the development of infection because the patient already had an active MRSA infection at that time. The patient was hospitalized for 12 days before developing catheter-associated bacteremia. The second bacteremia infection was classified as a community-acquired infection (CCI), defined as MRSA pneumonia with secondary bacteremia. The patient was not screened for MRSA and died a few days after admission. The three skin and soft tissue infections were classified as CCI. These patients had chronic ulcers of poor evolution with frequent admissions to the hospital. None of them were screened for MRSA on hospital admission during these episodes.

Whole-genome sequencing (WGS) was performed on isolates (both colonization and infection) from four residents who had a subsequent infection. This could not be carried out with the final resident because the strain that originated from his infection did not grow in the reseeding. Phylogenetic analysis demonstrated the genetic relationship between the colonization and infection isolates in three (75%) of the four residents studied (Figure 1). The discordant case was one of the skin and soft tissue infections. The results of molecular typing using MLST and were as follows: resident 1 (both strains were ST36), resident 2 (both ST5), resident 3 (both ST5) and resident 4 (ST8 for the nasal colonization strain and ST4697 for the infection strain). The results of the antibiotic resistance phenotype of the four representative STs were as follows: ST8 (mecA, blaZ, ermC, mphC, mrsA) ST36 (mecA, blaZ, ermA), ST5 and ST4697 (mecA and blaZ).

## 4. Discussion

In this multicenter point-prevalence study, a remarkable high colonization (28.1%) by MRSA was observed among LTCF residents in north Tenerife. These are residents with a high average age (80 ± 12) and with many basic pathologies and risk factors for acquiring infections, for this reason the study of colonization by multidrug-resistant organisms is very important for this type of population. The MRSA prevalence rate in our area has remained practically the same (25.8%) as a similar study carried out by our research group in 2014 [14]. In the literature, the prevalence of MRSA in LTCFs exhibits significant variability worldwide, showing large geographical differences [2,15,16,17,18,19,20,21,22,23]. The highest prevalence has been reported in studies from Asia, some of them reaching a rate of 65% [24,25]. The prevalence in Europe varies with an interquartile range between 4.4–19.6% [2]. In Spain there are few similar studies, also showing variability according to their geographical area, with a prevalence rate of 10.6% in studies from southern Spain [26] and 22.5% from Catalonia [27]. Differences in colonization rates may depend on several factors, including MRSA prevalence at the referring hospital, the geographic area, resident characteristics and infection control practices at the LTCF [28]. For this reason, it is important to carry out prevalence studies in LTCFs and to know the epidemiological situation in the local hospital area and to take preventive measures if necessary.

Unlike the study carried out in 2014 [14], there were no factors significantly associated with MRSA colonization, except having a history of previous MRSA. We observed that the prevalence of colonization in some LTCFs was very high, with more than half of the residents studied carrying MRSA. However, in two LTCFs we did not find any MRSA carriers, probably because these LTCFs had few beds and the number of residents recruited was low. In several previous studies, discordant results were also observed depending on the size of the LTCF; some of them reported a lower prevalence of MRSA in small LTCFs (below 35 or 40 beds) [26,28]. In general, the high global prevalence observed, including in residents with different characteristics and comorbidities, could suggest that we are in an endemic situation for MRSA in our hospital area.

During the two-year follow-up period, 5.6% of MRSA carriers developed infection, both intrahospital (related to health care) and extrahospital, with some of them being serious infections that caused the death of the resident. In the literature, a meta-analysis study [7] revealed that individuals colonized with MRSA exhibited elevated odds of developing infections in comparison to those who were not colonized. However, one limitation the authors reported was that none of the included studies conducted a comparison of strain types between colonizing and infecting isolates. In our study we carried out a phylogenetic analysis using WGS to compare the strains. In three of the four residents studied, the colonizing strain was the same strain that subsequently produced the infection. These results confirm that a patient colonized by MRSA can develop an infection from the same colonizing strain.

### Implications for Control Policies and Future Research Directions

Our findings highlight the importance of taking preventive measures for these types of patients, whether in hospitals, LTCFs or outpatient centers.

Given the high prevalence of colonization in LTCFs and that many of these residents require hospital admission (41.9%), it is necessary to take preventive measures in the hospital for patients admitted from LTCFs to avoid the development of infection and the spread of MRSA. Among the prevention strategies, screening at admission and having an alert system for patients with previous MRSA infections are highly recommended measures [8]. When the patients are positive in the screening, they should be put in strict isolation and receive a decolonizing treatment. Several recent studies have shown the benefits of a decolonizing treatment with nasal mupirocin and chlorhexidine baths in reducing the risk of developing MRSA infection [29,30]. Likewise, outpatient centers should be involved, and every time they attend patients with a history of MRSA, they should perform screening tests and decolonize those who are carriers.

On the other hand, LTCFs should also be recognized as a crucial setting in which to implement hospital infection control strategies [9]. Sasahara et al. [8] noted that the admission of new resident carriers of MRSA may be a significant factor contributing to the elevated prevalence of MRSA in LTCFs, rather than the spread via cross-transmission inside LTCFs. For this reason, screening at admission into LTCFs is one of the measures proposed [31], as well as contact precaution and regular screening. There is debate about the decolonizing treatment in LTCFs; some studies report a decrease in MRSA infections [32], however, other studies have not found any benefits [33]. Nevertheless, the implementation of these measures in LTCFs requires decisions at the national level. Therefore, implementing hand hygiene promotion and the specific use of gowns and gloves while caring for residents with wounds or medical devices seems to be the most optimal approach while awaiting guidance from the authorities [34,35].

Further studies using WGS are needed to better understand the epidemiology of MRSA and its behavior. In this context, a new research avenue has been opened, enabling the analysis of the actual risk factors for the development of an infection from the same strain in individuals carrying MRSA.

This study has several limitations. First, as the study was performed during the COVID-19 pandemic, we encountered significant challenges in gaining access to the centers for both sample collection and questionnaire completion, delaying study deadlines and preventing more residents from being included. Second, there could be more MRSA infections that we did not detect because LTCFs sometimes treat skin and soft tissue infections without sending samples for culture. Third, carriers with low levels of MRSA may have intermittent negative swabs and there may be false negatives, since only one sample was collected per resident. Fourth, the phylogenetic analysis could only be performed with eight isolates, which may affect the representativeness and generalization of the results. Despite these limitations, we consider that the number of residents included in the study, the follow-up period for colonized patients and the use of the WGS technique for the comparison of isolates add value to our results.

## 5. Conclusions

Our study reports a high prevalence of MRSA colonization among LTCF residents in the north of Tenerife. No risk factors were associated with MRSA colonization, finding ourselves in a probable endemic situation in our hospital area. WGS data allow us to confirm the epidemiological relationship between the colonizing and infectious strains. As a result, we can conclude that colonized residents can develop serious MRSA infections on hospital admission caused by the same strain of nasal colonization. These findings highlight the need to implement preventive measures in both LTCFs and hospitals to avoid the development of infection and the transmission of MRSA to other patients. Furthermore, more studies are needed to accurately quantify the risk of disease progression after colonization.

## Figures and Tables

**Figure 1 microorganisms-11-02842-f001:**
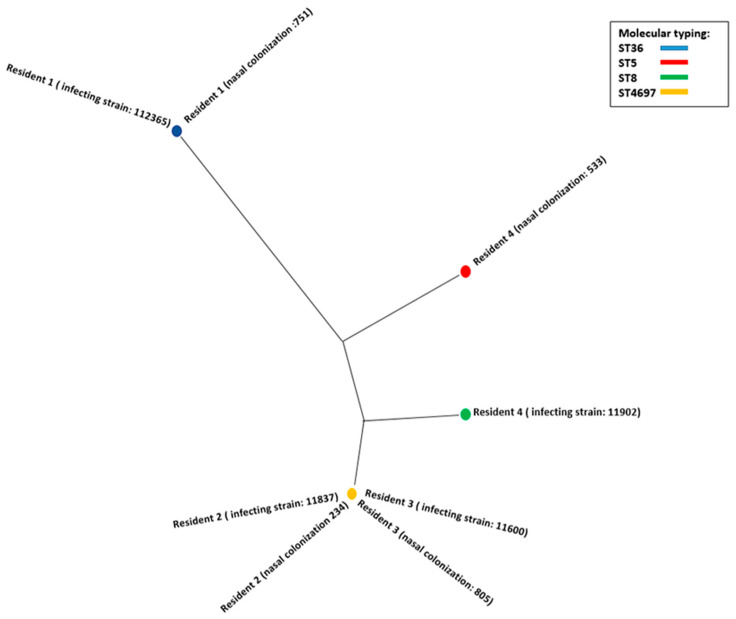
A graphical view of the phylogenetic tree of MRSA isolates. We can observe the genetic relationship between the colonization and infection strains of resident 1 (751-12365), resident 2 (234-11837) and resident 3 (805-11600). The strains of residents 2 and 3 are related to each other. However, the strains of resident 4 (533-11902) are not genetically related to each other.

**Table 1 microorganisms-11-02842-t001:** Distribution of residents by LTCF and the prevalence of MRSA carriers in each LTCF.

LTCF	No. Total Residents	No. Recruited Residents	Prevalence MRSA Carriers
LTCF-A	99	43 (43.4%)	6 (14%)
LTCF-B	99	71 (71.7%)	9 (12.7%)
LTCF-C	193	71 (36.3%)	19 (26.8%)
LTCF-D	102	70 (68.6%)	29 (41.4%)
LTCF-E	32	32 (100%)	9 (28.1%)
LTCF-F	86	78 (90.7%)	30 (38.5%)
LTCF-G	75	74 (98.7%)	21 (28.4%)
LTCF-H	20	20 (100%)	4 (20%)
LTCF-I	60	35 (58.3%)	6 (17.1%)
LTCF-J	130	127 (60%)	38 (29.9%)
LTCF-K	37	10 (27%)	0 (0%)
LTCF-L	600	74 (12.3%)	39 (52.7%)
LTCF-M	60	36 (60%)	5 (13.9%)
LTCF-N	48	23 (47.9%)	0 (0%)

**Table 2 microorganisms-11-02842-t002:** Comparison of potential predictive factors for MRSA colonization between resident cases and controls.

Variable	Total (*n* = 764)	Cases (*n* = 215)	Controls (*n* = 549)	*p* Value
Age (years)	80 ± 12	81 ± 11	79 ± 12	0.075
Female sex	511 (66.9)	146 (67.9)	365 (66.5)	0.707
Single room	94 (12.4)	26 (12.1)	68 (12.4)	0.915
Intrinsic risk factors				
Diabetes mellitus	259 (34.1)	76 (35.3)	183 (33.3)	0.561
Dermatitis	223 (29.3)	70 (32.6)	153 (27.9)	0.183
Peripherical vascular disease	178 (23.4)	52 (24.2)	126 (23)	0.687
Chronic kidney disease	58 (7.6)	16 (7.4)	42 (7.7)	0.938
Chronic obstructive pulmonary disease	96 (12.6)	29 (13.5)	67 (12.2)	0.611
Active infection	133 (17.5)	37 (17.2)	96 (17.5)	0.945
Urinary incontinence	563 (74.1)	161 (74.5)	402 (73.2)	0.554
Fecal incontinence	470 (61.8)	141 (65.6)	329 (60)	0.123
Extrinsic risk factors				
Dialysis	3 (0.4)	0	3 (0.5)	0.279
Central venous catheter	7 (0.9)	1 (0.5)	6 (1.1)	0.417
Urinary catheter	21 (2.8)	3 (1.4)	18 (3.3)	0.155
Feeding tubes	31 (4.1)	5 (2.3)	26 (4.7)	0.132
Previous antibiotic use *	315 (41.4)	93 (43.3)	222 (40.4)	0.439
Health requirement **				
High	368 (48.8)	98 (45.6)	270 (49.1)	0.884
Medium	265 (35.1)	75 (34.9)	190 (34.6)	
Low	121 (16)	34 (15.8)	87 (15.8)	
Prior hospital admission *	42 (5.5)	12 (5.6)	30 (5.5)	0.940
Length of hospital stay (days)	7 ± 2	11 ± 6	4 ± 1	0.258
Previous MRSA colonization ***	73 (9.6)	31 (14.4)	42 (7.7)	0.005

Note: Values are shown as mean (P5–P95) or *n* (%). * During the previous three months prior to study recruitment. ** According to local guides [13]. *** The resident has a history of MRSA carriage/infection (past year).

## Data Availability

The data presented in this study can be obtained upon request to the corresponding author.

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
