# Peer review of "The Impact of MRSA Colonization on Healthcare-Associated Infections in Long-Term Care Facility Residents: A Whole-Genome Sequencing-Based Study"

_microorganisms, 2023, doi:10.3390/microorganisms11122842_

Round 1
Reviewer 1 Report
Comments and Suggestions for Authors
This study aimed to determine the prevalence and risk factors for MRSA colonization among long-term care facilities (LTCF) residents of Tenerife (Spain), and to analyze the epidemiological relationship between colonizing and infecting strains using WGS. This work is meaningful in this field. However, revisions needed to be performed before accepted for publication.
Main Concerns:
(1) Staphylococcus aureus should be italic in all manuscript and references.
(2) Line 355, staphylococcus aureus should be replaced by Staphylococcus aureus.
(3) It was suggested to analyze the antibiotic resistance genes based on WGS.
(4) There are four ST types were isolated in this study. It was suggested to selected the representative strains among these four STs to analyze the antibiotic resistance phenotype.
Reviewer 2 Report
Comments and Suggestions for Authors
It was interesting to review this work, but I have some concerns before it can be published in Microorganisms:
General Comments:
The study titled "Is colonization by MRSA in long-term care facilities residents a real risk of developing a health care associated infection? A Whole-Genome Sequencing based study" examines the prevalence and risk factors of Methicillin-resistant Staphylococcus aureus (MRSA) colonization among residents in long-term care facilities (LTCFs) in Tenerife, Spain. The authors also investigate the epidemiological relationship between colonizing and infecting strains using Whole-Genome Sequencing (WGS). While the study addresses an important topic, there are several areas that require improvement and clarification.
Specific comments:
Title:
For me, the title needs to be improved and its structure containing question and separated phrases not appropriate in my point of view
Introduction:
The introduction lacks a clear contextual background and fails to provide a comprehensive rationale for the study. It briefly mentions that MRSA is a major public health concern and that LTCFs serve as important reservoirs for the microorganism. However, more information is needed to establish the significance of the study and its contribution to the existing literature. Additionally, the introduction should highlight the gap in knowledge that the study aims to address, especially regarding the genotypic comparison of colonizing and infecting strains. Providing a more comprehensive overview of previous research and the limitations of existing studies would strengthen the introduction.
Methodology:
The methodology section is concise and lacks sufficient detail. It would benefit from a more comprehensive description of the study design, including the selection criteria for LTCFs, the characteristics of the study population, and the data collection process. Additionally, the section should provide more information about the laboratory methods used for MRSA detection, confirmation, and strain typing. Details about the WGS technique, including the specific platform and protocols used, should be included. Providing this additional information would allow readers to better understand the study's methodology and replicate the research if desired.
Discussion:
Provide a more detailed discussion of the limitations of the study, including potential sources of bias and confounding factors.
Consider including a section on the implications of the study findings for clinical practice, infection control policies, and future research directions.
Conclusion:
The conclusion should be rewritten to provide a more concise summary of the study findings and their implications. It should address the prevalence of MRSA colonization among LTCF residents, the risk factors identified, and the epidemiological relationship between colonizing and infecting strains. The conclusion should also highlight the significance of these findings in the context of infection control strategies in LTCFs and potential measures to prevent MRSA transmission and infection. Overall, the conclusion should be more focused and clearly reflect the key outcomes of the study.
Overall, the study addresses an important research question regarding MRSA colonization and infection in LTCFs. However, it requires significant revisions to enhance clarity, provide more detailed methodology, and improve the conclusion's coherence and reflection of the study findings.
Comments on the Quality of English LanguageModerate editing of English language required
Reviewer 3 Report
Comments and Suggestions for Authors
Before proceeding with my comments, I want to express my empathy with the authors of the article. I can understand how frustrating it must be to carry out a sample of a large number of patients and miss opportunities for their follow-up when they attend outpatient services and valuable samples for research are not taken there. I imagine this is the reason (concatenated with COVID-19 epidemic) why this is not a full length article, and instead, the authors have opted for a Brief Report.
Line 158: Table 2. I have difficulties understanding some of the parameters expressed by the authors.
Regarding the ages of the sampled patients, "511" appears to be a numerical value that is not explicitly explained in the context. Without more context or information, it's challenging to interpret its specific meaning.
The value "66.9" is also unclear from the information provided. It doesn't represent the mean age, because the lowest confidence interval, given by the controls (79-12), is 67.
As for the representation of gender, only male gender is mentioned (it may be more appropriate to specify both) then ¿why is there a "+/- 12" interval?
Line 162: Could you explain how you calculate the values associated to “health requirement”?
Line 178: When referring to LTCF, it's not specified whether this includes retirement homes exclusively or other types of care facilities. This information is relevant because the result of 80 +/- 12 years of age, interpreted as a risk factor, encompasses almost all age groups of people residing in a retirement home.
Lines 194-200: The authors observe significant differences in the prevalences (0-50%) obtained in different LTCFs and mention their possible relationship with the number of residents. Why isn't this parameter studied in greater depth and considered among the factors in Table 2?
Lines 213-233: I agree with the ideas presented by the authors in these paragraphs. It could also be interesting, as highlighted by this study, to take into account the situation of MRSA in outpatient services, as no sampling is even done in patients with a history of MRSA.
Lines 259-260 and 261-262. Similar functions are attributed to different authors: “M.C.F and R.A.R carried out the collection of the samples and all the laboratory analyses” “MJ.R.R and Y.P.F participated in the collection and processing of the samples”
Line 260: A.A. Jaime should be changed by A.A.J.
Round 2
Reviewer 3 Report
Comments and Suggestions for Authors
Dear authors, a small clarification before finishing the publication.
Regarding Table 2: If 511 is the total number of female residents, wouldn't it be more appropriate to indicate the sexes as Female/Male instead of placing the number of men in parentheses? It gives the impression that 66.9 is an excerpt from 511. Additionally, how is it possible for the number of men to be a decimal?
Congratulations on the research.
